# Impact of urban land use on mean and heavy rainfall during the Indian Summer Monsoon

Renaud Falga[1] and Chien Wang[1]*

[1] Laboratoire d'Aérologie, University of Toulouse III – Paul Sabatier, Toulouse, France

*Email: chien.wang@aero.obs-mip.fr

**Abstract.** Northern India has been subjected to an intense urbanization since the middle of the twentieth century. The impact of such a drastic land-use change on regional weather and climate remains to be assessed. In this work, we study the impact of the modification of land-use – from vegetation to urban – on the Indian summer monsoon rainfall, as well as other meteorological variables. We use the regional meteorology model Meso-NH coupled with an urban module (the Town Energy Balance model) to perform month-long sensitivity simulations centered around Kolkata, the most urbanized area in Northeastern India. Paired simulations, one with and another without urban settings have been performed to identify the impacts related to urbanization through both thermodynamic and kinetic effects. We find that the perturbation induced by urban land-use enhances the mean rainfall over the model domain, principally by intensifying the convective activity through thermodynamical perturbation, leading to an increase of 14.4% of the monthly mean rainfall. In addition, the modeling results demonstrate that not only does the urban area act in general as a rainfall enhancer, particularly during nighttime, but it also induces the generation of a specific storm in one modeled case that would not have formed in the absence of the urban area. This storm initiation over the city was done primarily by urban terrain's disturbance of the near surface wind flow, leading to a surge of dynamically produced turbulent kinetic energy (TKE). The thermal production of TKE over the nighttime urban boundary layer, on the other hand, serves as a contributing factor to the storm formation.

## 1. Introduction

During the 20[th] century, India had witnessed major land-use and land-cover (LULC) modifications. Specifically, the urban areas and agricultural land-covers had been greatly expanded, especially between the 1950's and the 1980's, at an expense of forests and grasslands (Tian et al., 2014). Such modifications of LULC have had an impact on the local and regional climate, and in particular on the monsoon rainfall, including the extreme precipitation (Niyogi et al. 2018). In particular, potential environmental and climate impacts of urbanization have received a special attention of the scientific community in the past few decades (Qian et al., 2017). Urbanization-caused LULC change has been shown to have a significant influence on precipitation over, around, and downwind of the cities (Liu and Niyogi, 2019).

Urbanization can affect the local and regional climate through two distinct paths: (1) the emission of anthropogenic aerosols and, (2) the thermodynamical and kinetic perturbation induced by the modification of land-use. The former path, i.e., the urban aerosol emissions can modify the local surface energy balance by decreasing the amount of incoming solar radiation reaching the surface, and by increasing the amount of radiation absorbed by the boundary-layer atmosphere (e.g., Yu et al., 2006). Such emissions can also perturb the precipitation patterns through modification of cloud microphysical properties, also known as "aerosol indirect effect" (Twomey, 1976), which has significant impact around urban areas (e.g., Van Den Heever and Cotton, 2007). While the effect of anthropogenic aerosols on the

monsoon rainfall are believed to be important (e.g., Wang et al., 2009; Bollasina et al. 2011), in this study we focus on the second path, or the effects associated with the modification of LULC.

Urban land-use could alter the water cycle and precipitation through various effects on temperature, circulation, and moisture content. The most commonly recognized impact of urban areas on local climate is the temperature modification that they induce, in comparison to the rural background. This effect is known as the 'Urban Heat Island' (UHI), and has been fairly documented (e.g., Oke, 1982; Taha, 1997; Arnfeild, 2003). The reasons for the existence of this UHI include the decrease of surface albedo and thermal capacity as well as the increase in thermal conductivity of urban build-up in comparison to vegetation (Mohajerani et al., 2017). This means that the incoming solar radiation is stored more efficiently in human-made materials. The UHI is also known to be maximized during the night (e.g., Nakamura and Oke, 1988), due to the release of the heat stored in the buildings during the day. Furthermore, vegetation tends to have a cooling effect on it's surrounding environment through the evapotranspiration process. When liquid water evaporates, the energy required to break the hydrogen bonds between the water molecules is taken from the surrounding environment (surface and surrounding air), which has a local cooling effect. Therefore, the suppression of vegetation and thus associated evapotranspiration in urban areas can enhance the UHI effect (Chapman et al., 2018). Overall, the increase in surface temperature induced by the UHI has been shown to enhance the convective activity, and in turn the local precipitation (Han and Baik, 2008).

Another effect of urban areas on local meteorology is the modification of the wind patterns. The first known effect is the increase of surface roughness in the city, which induces a deceleration of the winds close to the surface. This can induce an increase in wind shear thus an enhancement of the turbulent mixing in the planetary boundary layer (Hildebrand and Ackerman, 1984), both have been shown to have the potential to invigorate thunderstorms (Weisman and Klemp, 1982). There has been some observational evidence that urban areas have an impact on thunderstorm paths as well as intensities (Niyogi et al., 2011; Shepherd et al., 2013). In fact, certain analyses using observations suggested that some thunderstorms might have been initiated by the urban area itself in the Atlanta region (Bornstein and Lin, 2000; Dixon and Mote, 2003). Shem and Shepherd (2009) attempted to verify these hypotheses using the Weather Research Forecast (WRF) model. They showed that the urban area did have a significant impact on the thunderstorm, but the results on storm initiation were inconclusive.

Nevertheless, the extent along with drivers of various potential effects of urban settings on water cycle proposed in previous works would depend on a wide variety of local factors, including meteorological and geographical features. For example, by acting as a low-level heat source, the UHI was shown to induce a low-level convergence of winds, creating a mechanically-forced "dome-shaped" convection over the city (Fan et al., 2017). However, this type of convection might not be sustained in an environment with strong background winds such as monsoonal trade winds. The presence of strong winds might result in an important heat transport that could reduce the strength of the UHI (Baik et al., 2001). Hence, the impact of urban land-use on precipitation in a large scale, highly energetic system like the monsoon has yet to be fully understood.

While the mean rainfall of the Indian Summer Monsoon (ISM) is believed to have decreased throughout the 20[th] century then reversed since the beginning of the 21[st] century (Jin and Wang, 2017), there is a general agreement among scientific community that the extreme rainfall events could have increased in India during the past monsoon seasons (e.g., Goswami et al. 2006, Dash et al. 2009; Roxy et al. 2017; Falga and Wang, 2022). Furthermore, some studies based on data analysis have linked this rise in rainfall extremes with the intense urbanization, by either comparing the pre-urbanization and post-urbanization trends (Shastri et al., 2015; Vittal et al., 2013), or by analyzing the precipitation at different urban locations (Kishtawal et al., 2010; Bisht et al., 2018, Swain et al., 2023). Mitra et al. (2011) identified certain observational evidences of the urban impact on the pre-monsoonal precipitations in Kolkata. However, there

have been only a limited number of studies using advanced models to examine the impact of urban land-use on monsoon rainfall. Lei et al. (2008) highlighted the need for suited parameterization of urban areas in models in order to accurately simulate the heavy rainfall rates of a particular storm case in Mumbai, and concluded about the important impact of urbanization on extremes. In addition to their observational study, Swain et al. (2023) also recently used the WRF model to show that the urbanization could induce an intensification of rainfall in the Bhubaneshwar region, located in Northeastern India. Li et al. (2017) also studied the projected impacts of LULC changes in the region, and found that an urbanization-increase scenario would lead to an increase in temperature and rainfall.

It is worth noting that urbanization appears as a local factor interacting with larger scale dynamic factors. For example, the Indian monsoon is known to be highly influenced by geographic particularities including orography on the West coast, but also in the Northern part of the country, where the Tibetan Plateau can influence aerosol concentrations, circulation, and precipitation (Liu et al., 2022). Thermodynamical considerations over the Indian Ocean can also explain rainfall sensitivity to large scale conditions, especially Northern India which is directly affected by the Bay of Bengal (e.g., Sheehan et al., 2023). Urbanization thus appears as a local perturbation with feedback to these large scale features.

In a previous effort, we have used a machine learning approach to show the correlation between the long-term trends of changes in LULC and those of extreme rainfall events (Falga and Wang, 2022). Specifically, urbanization trend becomes the key player among various types of LULC in the above correlation in northeastern India. Hence, in this study, we seek using a modeling approach to identify the causal relationship between urbanization and both mean and heavy rainfall during the ISM in a highly urbanized area of northeastern India: Kolkata. Instead of focusing on a single storm passing over the city, we simulate the whole month of July 2011 that covers several rainfall systems over the modeled area. This approach allows us to have a more comprehensive understanding of what happens after a storm passes over a city, and to determine if the rainfall enhancement by the urban areas is consistent among different cases. The lengthy simulations also enable us to highlight some key features of the urban perturbations in Kolkata, such as the diurnal cycles of UHI, surface fluxes, and rainfall perturbations.

In the following, we firstly present the model setup and the data used in our modeling. Then, we discuss several general characteristics of the perturbation of urban settings and its consequent effects on rainfall in part 3 of this paper. In part 4, we present the detailed analysis results that show how urban settings can even induce the initiation of a nighttime storm, in addition to enhancing the rainfall. We then discuss the potential influence of the synoptic scale conditions on rainfall modification in part 5. The last section of the paper summarizes our findings.

## 2. Model setup and data

### 2.1 The model

We utilize a regional meteorological model, the Meso-scale Non-Hydrostatic model or Meso-NH (Lac et al., 2018) in this study. Meso-NH model has been developed jointly by Laboratoire d'Aérologie (UMR 5560 UPS/CNRS) and the French National Centre for Meteorological Research (CNRM, UMR 3589 CNRS/Météo-France). The model includes a two-moment microphysical module for Liquid, Ice, Multiple Aerosols, or LIMA (Vié et al., 2016) to predict evolutions of various hydrometeors and aerosols. The European Center for Medium-Range Weather Forecasts (ECMWF) radiation scheme is included to simulate short- and long-wave radiation (Hogan and Bozzo, 2018). The simulations use an eddy diffusivity mass flux scheme (Pergaud et al. 2009) to parameterize the shallow convection. Deep convection and associated physio-chemical processes in our configuration are explicitly resolved.

In this study, the Surface Externalisée (SURFEX) scheme is used for surface fluxes and land-surface interaction

processes (Masson et al., 2013). SURFEX is a surface modeling platform developed by Météo-France, in which each model grid is separated into four tiles: Nature, Sea, Lakes, and Town, and each one of these tiles uses a different parameterization. In particular, the Town Energy Balance (TEB) model (Masson, 2000) is used on the 'Town' tile to calculate the energy and water fluxes between the urban grids and the atmosphere. The TEB is an urban canopy model that takes an important number of physical processes into account, including building-scale processes, while also managing to maintain an accurate representation of the 3D geometry of the city. Three different surface types are defined: roads, streets, and walls, and distinct energy budgets are computed for each of these surface types. Some geometrical parameters like building height, width, or orientation are constant within a grid cell. The TEB has been demonstrated to be able to accurately simulate thermodynamical effect of urban areas like the UHI (Lemonsu et al., 2002), or the modification of surface energy balance (Pigeon et al., 2008).

## 2.2. Model domain and configuration

The Meso-NH model has been configured to have a horizontal resolution of 900 meters and 62 stretched vertical layers, with a vertical spacing of 10 meters near the surface and gradually maximized to 1000 meters at the top of the model domain. Such a spatial resolution configuration allows us to resolve the deep convection directly without parameterization. The model domain covers an area of 360x360 km² centered around Kolkata. In order to assess the impact of the urban area on its immediate surroundings, we have also defined a smaller area in the proximity of Kolkata (noted as 'Kolkata area' in the rest of the paper), covering an 80x80 km² area. The spatial coverage of the domain as well as the urban fractions used in the 'Urban' run, and the boundary used to delimit the Kolkata area are shown in Fig. 1.

We have performed a set of two sensitivity simulations using Meso-NH coupled with town energy balance model. The first simulation (hereby referred to as 'Urban' run) uses recent, realistic land-use data obtained from the ECOCLIMAP-II database (Faroux et al. 2013). In the second control simulation (referred to as 'No-Urban' run), the urban land-use settings are replaced by the local surrounding vegetation, i.e., grassland. The configurations of these two simulations are otherwise identical.

The modeled period in all simulations is the whole month of July 2011. The choice of the year of 2011 was made based on both climatological consideration and the availability of supporting data. This year is found to be an overall 'average' year over the past decades in terms of extreme rainfall events during the Indian summer monsoon seasons, meaning that the frequency as well as the intensity of rainfall extremes were close to their climatological mean values over the recent decades (Falga and Wang, 2022).

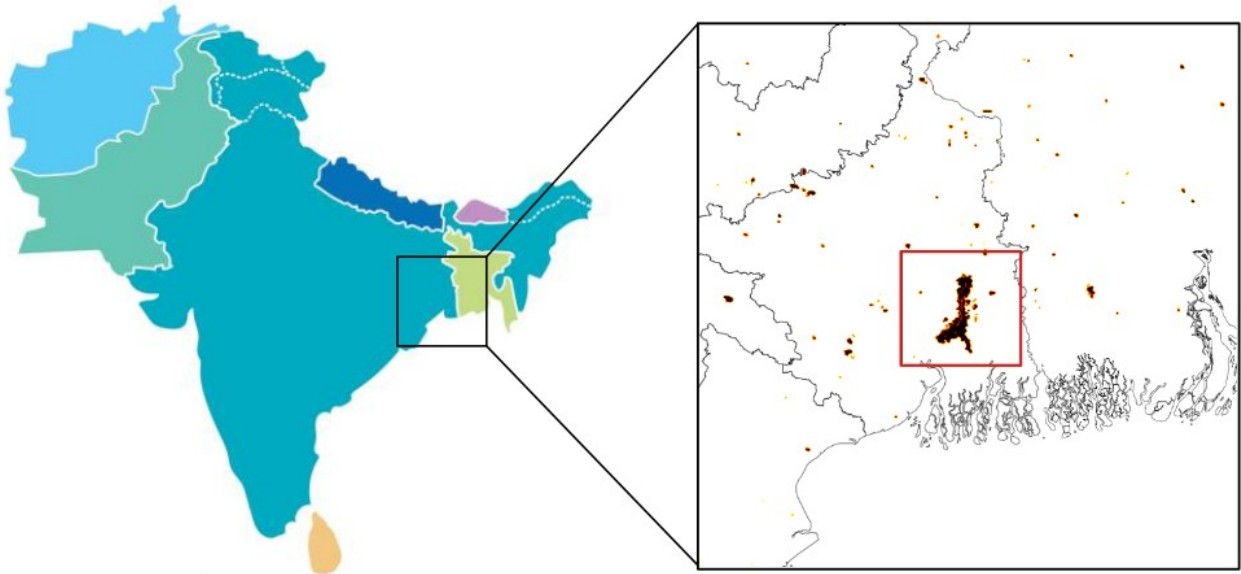

**Figure 1** – Model domain used in the simulations. The urban fractions in the domain (in dark colors) as well as the boundary defining the Kolkata area (red square) have been plotted. In the reference run, the urban areas have been replaced by the surrounding vegetation, i.e., grassland. The whole domain is square-shaped and the length of one side is 360 km.

160

As initial and lateral boundary conditions, we use the ECMWF Integrated Forecasting System (IFS) data (IFS Documentation CY47R3, 2021). The boundary conditions are introduced through time sequence coupling files every six hours. The 12-hourly prescribed Sea Surface Temperature (SST) values, obtained from the ERA5 reanalysis data (Hersbach et al., 2020) are also used to drive the simulations. As the aerosol effect on precipitation is not a research focus here, we have prescribed constant aerosol concentrations in all simulations (300 aerosols/cm³ for each of the three Cloud Condensation Nuclei or CCN modes).

### 2.3. Model comparison with ERA5 reanalysis data

When comparing our modeled rainfall with the ERA5 reanalysis data, we find that the model has largely reproduced the evolution of domain-averaged hourly rainfall time series in ERA5 (Fig. 2). Specifically, the two major heavy rainfall periods shown in the reanalysis data, happening from July 1 to July 5, and from July 18 to July 22, respectively, are both simulated successfully by our model. While precipitation intensity in the first period seems to be somewhat overestimated by the model, the quantity in the later one, however, matches ERA5 data quite well. Note that the

discrepancies between the model and reanalysis could come from either the difference in spatial resolution between our model and ERA5, or the simplified representation of aerosols in our model.

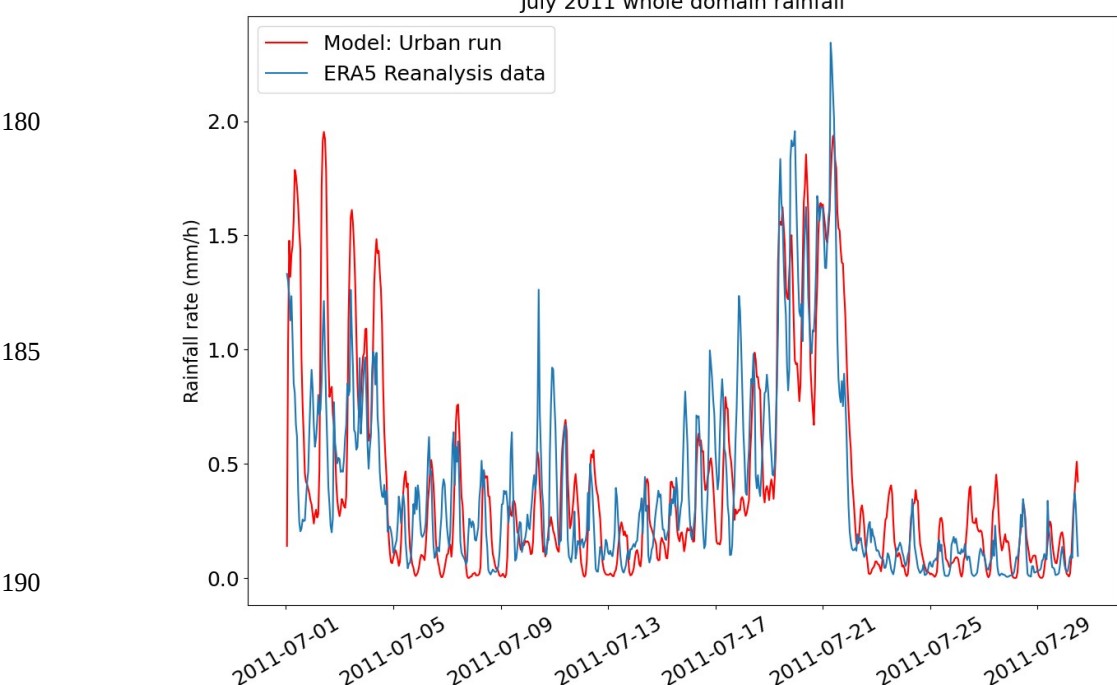

**Figure 2** – Domain-averaged rainfall time series of our modeled Urban run (in red) and the ERA5 reanalysis date (in blue)

Note that the intention of this study is not to simulate exactly the observed rainfall, but rather investigate the sensitivity of the simulated precipitation to land-use modification. Therefore, we need to insure that the simulated rainfall is able to reproduce large-scale features rather than exact rainfall at specific locations. ERA5 reanalysis provides consistent and high resolution data for the whole modeled month over the model domain, the comparison with ERA5 thus appears suited for our study.

## 3. Urban impacts on monthly meteorological features

### 3.1  Characteristic urban impacts on physical features

The month-long simulations provide us with an opportunity to understand not only how certain physical features are affected by urban setting in each individual day, but also increases the statistical significance of such effects throughout the entire modeled time frame. As mentioned previously, certain characteristics of the urban impact on the regional meteorology and climate depend greatly on local factors. Hence, the effects of Kolkata on some local meteorological features and their diurnal cycles have been studied first (Fig. 3). For instance, we find that the Kolkata area induces a clear perturbation in the diurnal cycles of surface heat fluxes, both sensible and latent, as shown in Fig. 3 (a). The differences between Urban and No-Urban run are most evident during the day for both fluxes, with the largest difference occurring at noon, when the incoming solar radiation reaches its maximum. Specifically, surface sensible heat flux is higher in the Urban run, due to a lower albedo and heat capacity, and higher heat conductivity of the urban area, leading to a higher surface temperature during the day. On the other hand, the reduced vegetation in the Urban run induces a lower evapotranspiration, which explains the higher latent heat flux in the No-Urban run. While evaporating into vapor, the water takes energy from the surrounding environment, thus inducing a local cooling. However, the temperature of the water does not change, and the energy required for evaporation is stored within the water vapor in the form of latent heat. This explains why the latent heat flux above vegetation is higher than above dry surfaces.

Therefore, a weaker latent heat flux is an indication of lower evaporation processes, so the UHI effect is further
strengthened by such a reduction of latent heat flux. Despite the opposite signs attached to the differences in sensible
versus latent heat flux, both favor a higher temperature in the Urban run, and their combined effects cause an intense
positive surface temperature difference between Urban and No-Urban run that can reach over 10 Kelvin at noon (not
shown here).

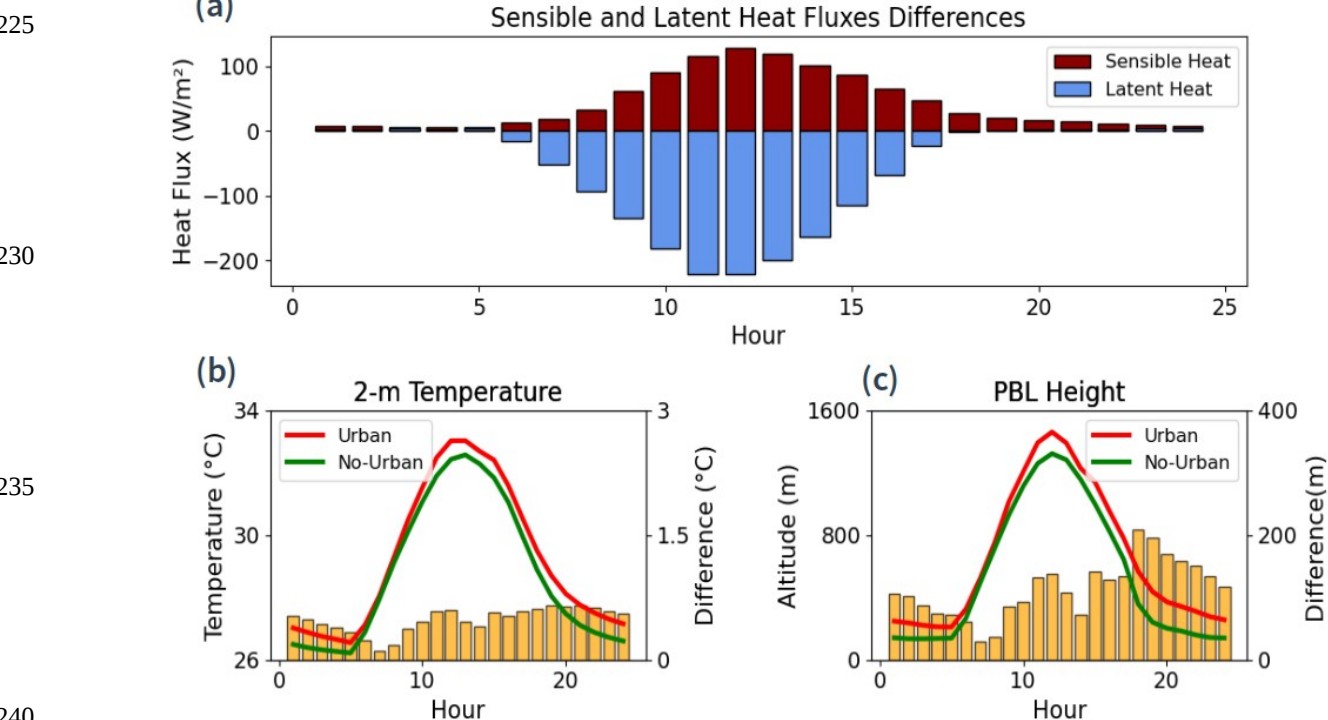

**Figure 3** – Monthly-mean diurnal cycles of (a) surface sensible and latent heat flux differences (Urban – No-Urban),
(b) 2-m temperature, and (c) PBL height. The solid lines in (b) and (c) show the actual diurnal cycles, and the orange
bar plots show the associated differences (Urban – No-Urban). In (a), only the differences are shown. All these
variables are calculated within the grid cells where Kolkata is located.

Whereas the surface temperature difference can reach such high values, the variable most often used to calculate the
UHI intensity is the difference in 2-m temperature. This difference is often relatively smaller, well reflecting the actual
temperature difference experienced between the city and rural background. In our simulations, the UHI, defined as the
difference in 2-m temperature over Kolkata between Urban and No-Urban run, is on average in its highest value of 0.65
K at 19:00, or the end of the day (Fig. 3 (b)). After 19:00, despite the air temperature dropping in both simulations, the
UHI remains relatively constant until midnight, with UHI value still as high as 0.55 K at midnight. This is due to the
physical properties of urban build-up, which allows heat to be stored during the day, and released during the night. The
UHI intensity starts to decrease more rapidly after midnight before reaching its minimal value of about 0.1 K at 7:00,
when most of the heat stored during the previous day was released.
The diurnal cycle of UHI shows a similarity with that of the Planetary Boundary Layer (PBL) height (Fig. 3 (c)), as
both are directly affected by the incoming solar radiation. The PBL starts to develop in the morning when the vertical
mixing begins to strengthen, and stabilizes at sunset. By enhancing the surface temperatures, the urban areas cause a
rise of the vertical temperature gradient and buoyancy, which further increase turbulent mixing as well as local
convective activity and thus PBL height. The heights of PBL in both Urban and No-Urban run are peaked at 12:00
(Fig. 3 (c)). Then, PBL height in the No-Urban run quickly decreases as the vertical air mixing weakens with the

reduction of incoming solar radiation, while remaining relatively higher in the Urban run. In fact, the difference in PBL height between the Urban and No-Urban run is the highest at the end of the day around 18:00. Just like the UHI, the difference in PBL height remains evident throughout the night, and the nighttime PBL is as much as twice deep in the Urban than No-Urban run, suggesting that vertical mixing within PBL remains relatively active during the night over the urban area.

### 3.2 Characteristic urban impacts on rainfall statistics

To assess general characteristics of the urban impact on the rainfall statistics, the averaged diurnal cycles of rainfall within Kolkata and surrounding area (the red square shown in Fig. 1) over the modeled month, as well as the monthly mean rainfall in both simulations have been calculated (Fig. 4). We have chosen to display the precipitation results in this area of 80 km by 80 km, as the effects of urban areas on rainfall are believed to be quite localized. In their meta-analysis, Liu and Niyogi (2019) showed that the rainfall modification happened on average about 50 km downwind of the city, and approximately 40 km on average around and upwind. Therefore, the size of the Kolkata area appeared to be suited and in agreement with previous studies for analysis on rainfall modification. Note that when looking at rainfall modification in the whole domain, the differences remain fairly light, as the urban effect outside of the Kolkata area appear to be negligible. Note also that this is not due to a potential rainfall reduction occurring outside the Kolkata area that would offset the enhancement in the Kolkata area, but rather due to the absence of significant differences anywhere else in the domain, thus dimming the effect of Kolkata on the whole domain rainfall.

The diurnal cycles of rainfall show the same type of behavior as the 2-m temperature described previously, but with a peak occurring a few hours later (see Fig. 4). The Urban and No-Urban simulations have a maximum rainfall in the afternoon at 17:00 and 16:00, respectively, meaning that more intense precipitation happens during afternoon when the convection is strong. However, while the precipitation amount decreases throughout the night in the No-Urban run, it remains relatively high in the Urban run, such that the maximum precipitation difference between the two happens at 2:00. At this time, the average rainfall is 86% more intense in the Urban than No-Urban simulation. In the previous subsection, we found that the nighttime UHI induced higher nocturnal PBL, which indicates less atmospheric stability in the Urban run and favors the possibility of rainfall occurrence. The analysis of diurnal cycles of rainfall therefore further confirms this nighttime instability in the urban boundary layer, as the rainfall enhancement by the urban area is much more evident during the night.

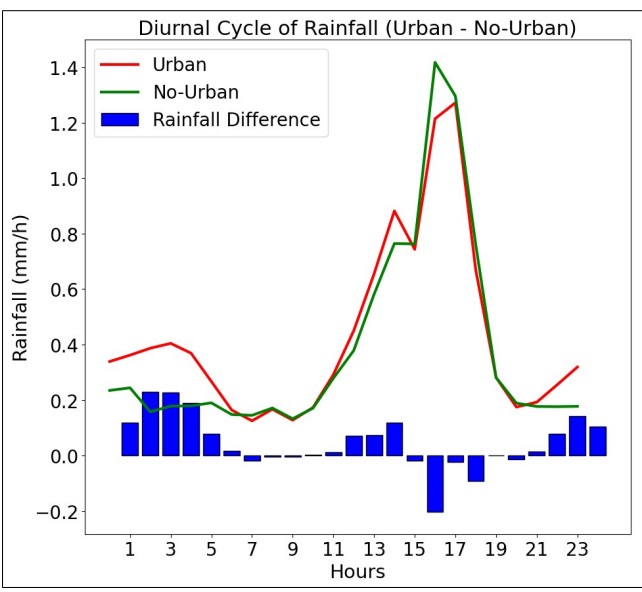

**Figure 4** – Rainfall diurnal cycles. The diurnal cycle of rainfall for the Urban (No-Urban) simulation is shown in red (green). The blue bar plot correspond to the differences (Urban – No-Urban) of these diurnal cycles.


From monthly mean rainfall (Fig. S1), defined as the time-average rainfall over the simulated month at each grid cell of the Kolkata area, we find that the mean value in No-Urban simulation (Fig. S1 (a), right panel) is about 2.30 mm/6h, while in the Urban simulation (Fig. S1 (a), left panel), the mean rainfall is significantly higher at 2.63 mm/6h, corresponding to an overall monthly increase of 14.4 %. This rainfall enhancement by the urban setting appears to be

even more intense over the city itself. When looking at the model grids within Kolkata, the mean rainfall is 22.8 % higher in the Urban than No-Urban run. The Probability Density Functions (PDF) corresponding to these two rainfall maps (Fig. S1 (b)) show a clear shift of the Urban rainfall towards the higher value side.

In order to better understand the time evolution of precipitation and the underlying processes involved in the urban-caused rainfall modification, we have calculated the time series of rainfall for both simulations, averaged in the Kolkata

area (Fig. 5; note that the values shown are six hour-accumulated rainfall, not the hourly values). During the month of July 2011, three distinct heavy rainfall periods have been identified (Fig. 5, marked by black rectangles). During the first and third periods, lasting from July 1[st] to July 5[th] and from July 18[th] to July 22[nd], respectively, the mean rainfall over Kolkata was significantly enhanced due to the urban settings, reflected from an average increase of 14.9% in the Urban than No-Urban run during the first period and 15.0% during the third. As will be further discussed in part 5, both these

two periods were associated with a synoptic scale low-pressure system. On the other hand, during the second period (July 10[th] to July 13[th]), rainfall is much less intense than those of the two other periods, and both rainfall reduction and enhancement due to urban settings appeared consecutively in the area, though the overall rainfall modification is almost negligible (a +0.5% increase). Therefore, we will focus on the first and third periods in the following discussions.


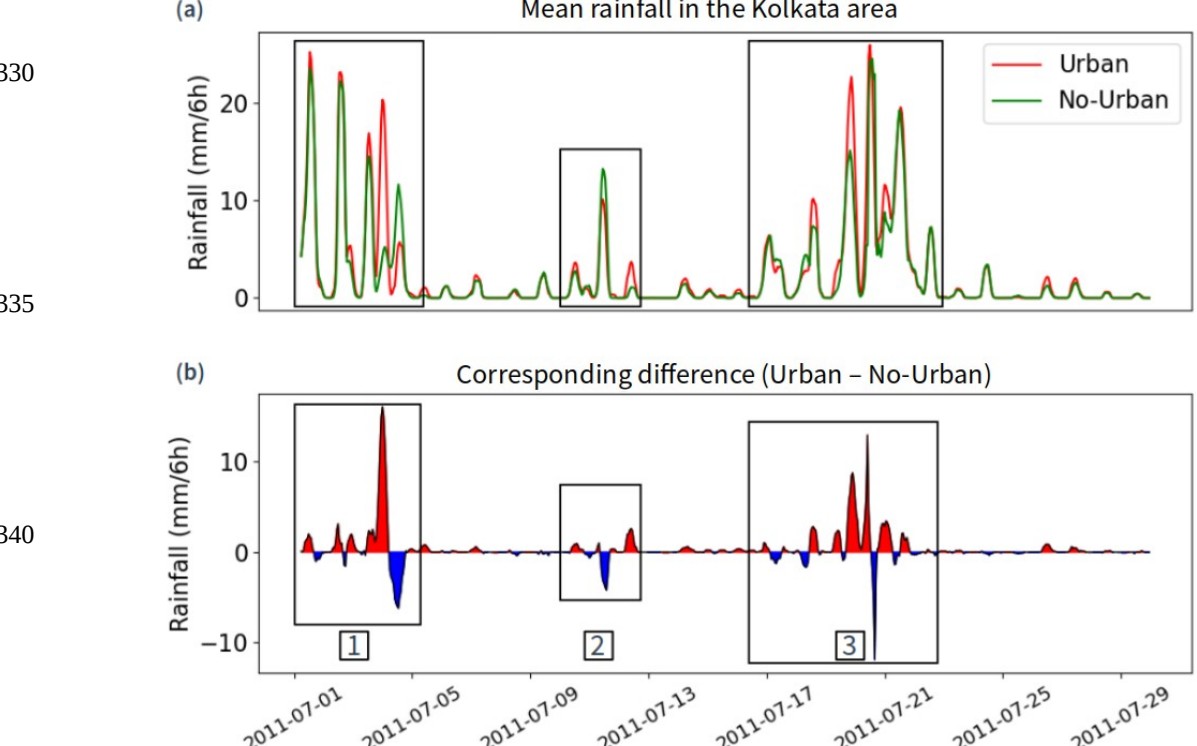





**Figure 5** – Time series of mean rainfall, averaged in the Kolkata area. The top panel (a) shows the time series of precipitation and the corresponding differences (Urban – No-Urban) are plotted in the bottom panel (b). The red (blue) parts in (b) correspond to times when the mean rainfall is higher in the Urban (No-Urban) run. Three distinct heavy rainfall systems passing over the area have been highlighted and numbered for clarity (black rectangles).


We have also assessed the modification induced by the urban land-use on two different extreme precipitation events indicators: the 99th percentile and the maximum rainfall. The former corresponds to the hourly 99th percentile of the 8100 grid cells rainfall values in the Kolkata area. For the latter, we keep only the hourly maximum value in the area. The time series of the differences (Urban – No-Urban) are shown in Figure S2. The 99th percentile indicator in the

Urban run shows an average increase of 7.4% in comparison to the No-Urban run. For the maximum rainfall, the average increase reaches 14.8%.

## 4. The in-depth analysis of an urban-initiated heavy rainfall event

Most of the time during the simulated month, the urban area acts as a 'rainfall enhancer', meaning that the

precipitation in the Urban run is simply higher than that in the No-Urban run (Fig. 5). However, the fourth rainfall peak in the first heavy rainfall period (see the first black box in Fig. 5) appears to be specifically interesting: the mean rainfall is very intense in the Urban run while close to zero in the No-Urban run. This corresponds to the highest positive rainfall difference between these two runs (Fig. 5 (b)), which starts on July 3rd 22:00 and ends on July 4th at 10:00 local time. In comparison, the three preceding rainfall peaks all come from diurnal convective precipitation: the

incoming solar radiation causes the convection and rain to be initiated during the day, but precipitation stops at nighttime when the boundary layer becomes stable. Indeed, the 6-hour accumulated rainfall maxima all happened at 18:00 for these three cases. This type of diurnal convective rainfall is slightly higher in the Urban run, with the rainfall maximum for all three peaks about 10% higher. This further confirms that the urban area does increase the convective activity, as was shown in section 3. However, note that even though the rainfall is enhanced by the urban area, the

heavy precipitation is present in both of the simulations, which shows that during these first three days, the main drivers of the rainfall were other factors such as the solar radiation as well as the important moisture availability (this last point is further discussed in section 5). On the other hand, the fourth rainfall peak represents a system that was initiated in the proximity of Kolkata in the Urban run at the end of the third day, with most of the rain falling during the night in the Kolkata area. This storm would not have formed if the city was absent since the rainfall in No-Urban run during the

corresponding time was extremely light. In fact, the rainfall is more than three times more intense in the Urban run than in the No-Urban run (about 240% more intense over the whole night). When compared to the previous 10% enhancement of the rainfall, the difference between the different peaks becomes evident. Therefore, the processes involved in the rainfall modification at this time differ from the three previous rainfall peaks. Consequently, while we refer to the rainfall modification as "enhancement" or "reduction" during the rest of the simulation, this fourth rainfall

peak corresponds to an "initiation" of a heavy rainfall event by the city. Investigating the physical processes involved would enable us to better understand how this nighttime storm formed over Kolkata due to urban settings.

### 4.1. Storm initiation and consequent evolution

Figure 6 shows the storm in its beginning, evolution, and propagation stage (note that the left and right panels

correspond to the hourly rainfall in the Urban and the No-Urban run, respectively). At 22:00, before the storm reaches the city, a precipitation zone already appears slightly west of Kolkata in the Urban run, but nothing happens in the No-

Urban run (Fig. 6, (a) and (e)). At this point, the precipitation remains relatively light (inferior to 10 mm/hour). This moderate cloud system then propagates eastward before reaching Kolkata around midnight, or two hours later. Three hours later at 1:00 (panels (b) and (f)), the rainfall has greatly intensified in the Urban run when the cloud system reached Kolkata, with hourly rates increasing up to more than 100 mm/hour (panel (b)). In contrast, as seen in panel (f), this storm is absent in the No-Urban run. After leaving the urban proximity, at 5:00 (panels (c) and (g)) and then 10:00 (panels (d) & (h)), the storm propagates south-southwestward in the Urban run, before its dissipation in the Bay of Bengal. The approximate storm path is shown in each panel of the Urban simulation in red dashed lines. In contrast, there is no corresponding system developed in the No-Urban run.

We have also compared the detailed thermodynamic and dynamical structures of both Urban and No-Urban simulation for the moments just prior to and also after the initiation of the storm. The relatively straight trajectory of this storm allowed us to select a cross section for analyzing certain three-dimensional variables aligning almost perfectly with the storm path.

Figure 7 shows the distributions of the equivalent potential temperature (noted $\theta_e$) anomalies, wind patterns, and the Total Condensed Water (TCW) along the storm path over the selected cross sections (see Fig. 6, red dashed lines) for both simulations at three different time steps: 22:00, 2:00 and 5:00. $\theta_e$ is a useful variable for studying the atmospheric thermodynamics and stability under a humid environment like the ISM season. It is similar to the standard potential temperature except that it includes the temperature change brought by the release of latent heat through condensation, thus providing information about both temperature and humidity content of the atmosphere. These cross sections shown in Fig. 7 therefore provide us with an insight on the evolutions of circulation, humidity, and atmospheric stability immediately before, at, then after the storm initiation. Note that the $\theta_e$ anomalies here are calculated at each grid of the cross section by subtracting the vertically averaged $\theta_e$ value of the grid cell to the values in the column. Hence, a positive anomaly at the surface means that the surface value of $\theta_e$ is higher than its vertical average and indicates that $\theta_e$ decreases with height, which is an indicator of instability.

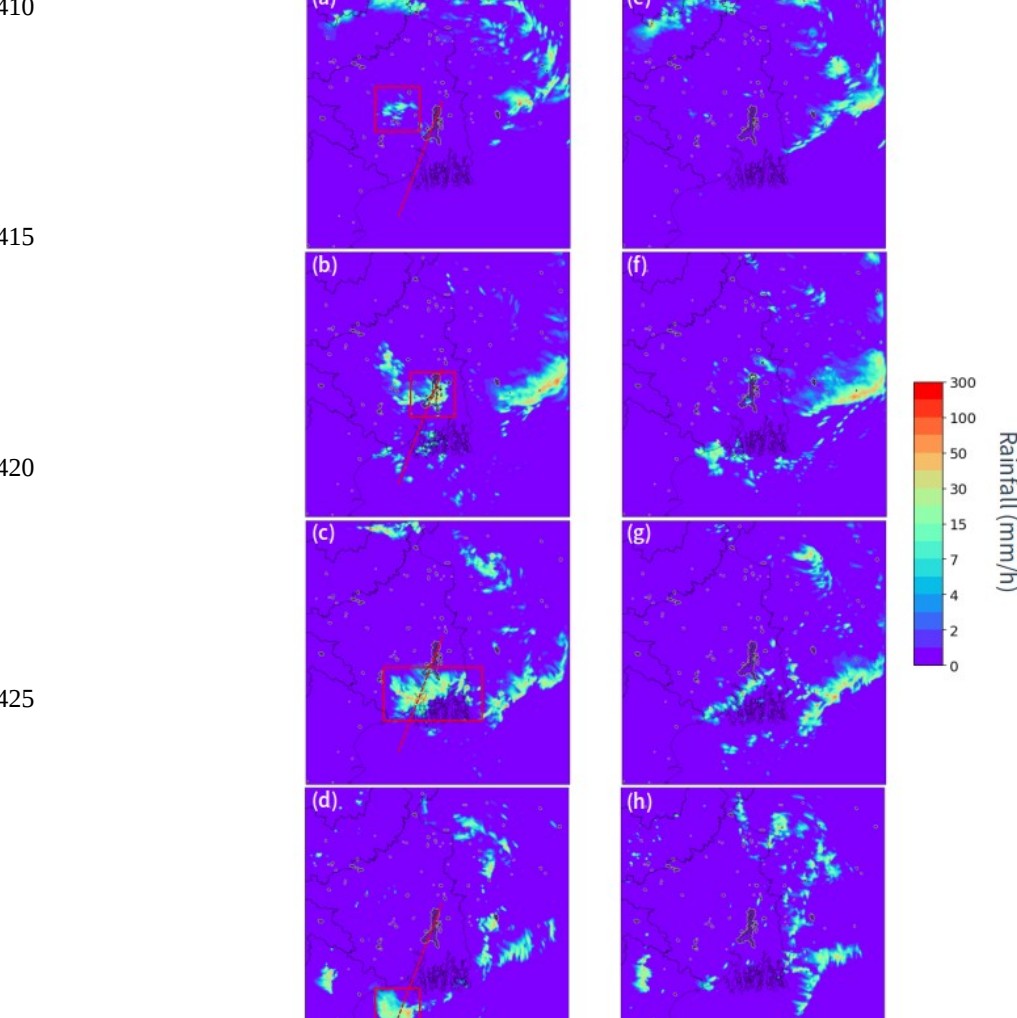

**Figure 6** – Storm development and propagation. Panels a, b, c and d (e, f, g and h) correspond to the hourly simulated rainfall in the domain respectively at times 22:00, 1:00, 5:00 and 10:00 in the Urban run (No-Urban run). The delimitation of the storm initiated in the Urban run is shown in red (left panels). The red dashed line shows the trajectory of the storm, which is where we calculated the Fig. 7 cross section. Note that the color scale here is not linear.

At 22:00, just before the storm was initiated, from near the surface up to an altitude of approximately one kilometer, strong and steady winds blow from the ocean to the land (from south to north), which is a typical wind pattern occurring during the summer monsoon season (Fig. 7 (a) and (b)). The corresponding strong positive $\theta_e$ anomalies suggest that these southerly winds have brought moist and unstable air to the continent. Then three hours later, when the humid and unstable flow reaches the city (black dashed lines), the positive $\theta_e$ anomaly propagates upward in the Urban run, the air is lifted and the convection is initiated in this simulation (Fig. 7 (c)). This convection is rather deep, as the updrafts eventually reach an altitude of about thirteen kilometers. The shape of the forming storm can be seen from the contour of total condensed water. After the storm passes the city, a mid-tropospheric return flow leads the storm moving southward, as seen on the bottom left panel (Fig. 7 (e)). Right after the passage of the storm, the instability of the atmosphere has been reduced, reflected from the weakening positive anomalies in the lower atmosphere and a close to zero vertical gradient of $\theta_e$, which is an indicator of a well-mixed atmosphere. In contrast, despite the same moist and unstable low-level southerly flow reaching Kolkata at 22:00 in the No-Urban run (Fig. 7 (b)), no updrafts have been developed thereafter at 2:00, and the flow just continues to propagate northward (Fig. 7 (d)). This suggests that the perturbation induced by the urban land-use in the PBL has allowed the deep convection to be initiated in the Urban run, while the analysis of $\theta_e$ and wind patterns in a larger scale indicates that the warm and moist air mass was advected from the Bay of Bengal to the land by the low-level monsoonal winds. The processes of this surface perturbation will be further analyzed in the Kolkata area in the following subsection to determine both kinetic and thermodynamical factors that have played an important role in this storm initiation.

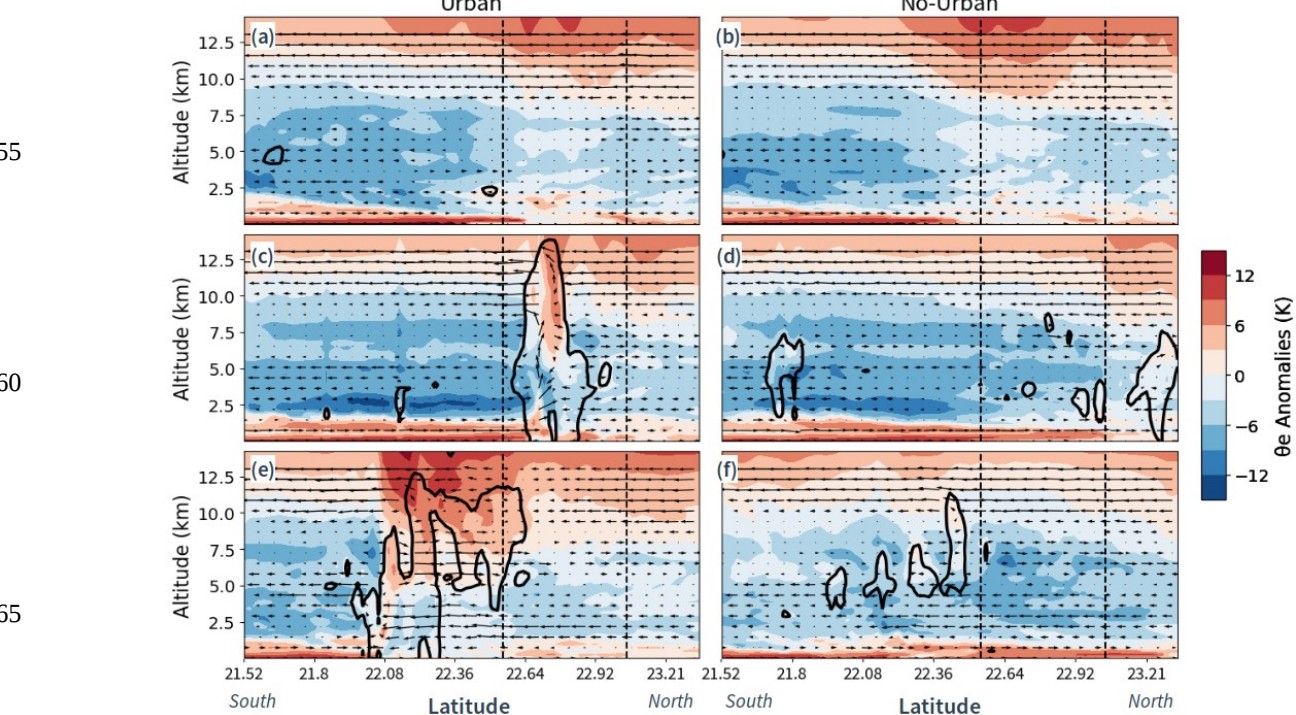

**Figure 7** – Cross section of the equivalent potential temperature anomalies. The panels (a), (c) and (e) show the anomalies along the storm path in the Urban run, at three different time steps: 22:00, 1:00 and 5:00. The panels (b), (d) and (f) show the same time steps but for the No-Urban run. The projected two-dimensional wind field is plotted on the

cross sections and we also plotted the TCW contour (TCW = 0.3g/kg). The right part of the figure correspond to the north end of the cross section and the storm propagates southward in the Urban run (towards the left of the figure). Kolkata is delimited by the black vertical dashed lines. The $\theta_e$ anomalies here are calculated at each grid of the cross section by subtracting the vertically averaged $\theta_e$ value of the grid cell from the values in the column, therefore giving us insights on the vertical profiles.

## 4.2. Evolution of turbulent kinetic energy and storm initiation

As convective storms usually develop in a sub-hourly timescale, the hourly output set originally in our simulations are thus not ideal for performing a detailed analysis. Therefore, we have conducted an additional set of two 28-hour (i.e., Urban and No-Urban, respectively) simulations between July 3rd – 12:00 and July 4th – 16:00, with an output every five minutes, initiated using the outputs of July 3rd – 12:00 from the first set of two simulations.

As mentioned in the previous subsection, the deep convection was initiated in the proximity of Kolkata. It developed from an existing moderate rainfall system that formed slightly West of the city around 22:00 (Fig. 6 (a)). This system then propagated eastward and reached Kolkata shortly before midnight. The more frequent outputs have clearly revealed that the rainfall system slowed down when arriving in the proximity of the urban area, as if the city acted like a barrier. The system arrived in the proximity of the city and started to slow down at about 23:30. Along with this deceleration, there was a sharp increase in Turbulent Kinetic Energy (TKE) at the surface, with a maximum value at 23:55 (Fig. 8 (a)). In Meso-NH, the evolution of the TKE e is calculated by closing the prognostic turbulence kinetic energy equation:

$$\frac{\partial e}{\partial t} = -\frac{1}{\rho_{dref}} \frac{\partial \left( \rho_{dref} e \overline{u_i} \right)}{\partial x_j} - \overline{u_i' u_j'} \frac{\partial \overline{u_i}}{\partial x_j} + \frac{g}{\theta_{vref}} \overline{u_3' \theta_v'} + \frac{1}{\rho_{dref}} \frac{\partial}{\partial x_j} \left( C_T \rho_{dref} L e^{1/2} \frac{\partial e}{\partial x_j} \right) - C_d \frac{e^{3/2}}{L}$$

Where u is the wind, $\theta_v$ is the virtual temperature, $\rho_{dref}$ is the volumetric mass of dry air for the model reference state, L is the characteristic length scale of the sub-grid eddies. In this equation, the terms on the right-hand side of the equation respectively represent the advection of TKE (first term), the shear production (second term), the buoyancy production (third term), the diffusion (fourth term), and the dissipation (fifth term). Therefore, there are two distinct production sources of TKE: thermal (the third term) and dynamic (the second term). The former corresponds to the buoyancy flux, i.e. fluctuations induced by thermodynamical instability, while the latter refers to the effects related to friction and wind shear. Evidently, the above identified burst in TKE was dynamically produced since the thermal contribution to TKE was negative (Figure S3). This increase in surface TKE was accompanied by an increase in cloud top height and the triggering of deep convection, with the cloud system reaching thirteen kilometers of altitude at the moment of the burst, right above the location of the TKE sudden increase (Fig. 8 (b)). During that time, the rainfall intensified, and the intense precipitation over the city appears to have been initiated at this place, with heavy rainfall appearing just 10 minutes after the burst of TKE (Fig. 8 (c)), before spreading all over the city (Fig. 8 (d)). This result highlights the important role of the surface roughness of the city in disturbing wind pattern near the surface, creating a burst of TKE in the modeled case, and thus inducing the storm development.

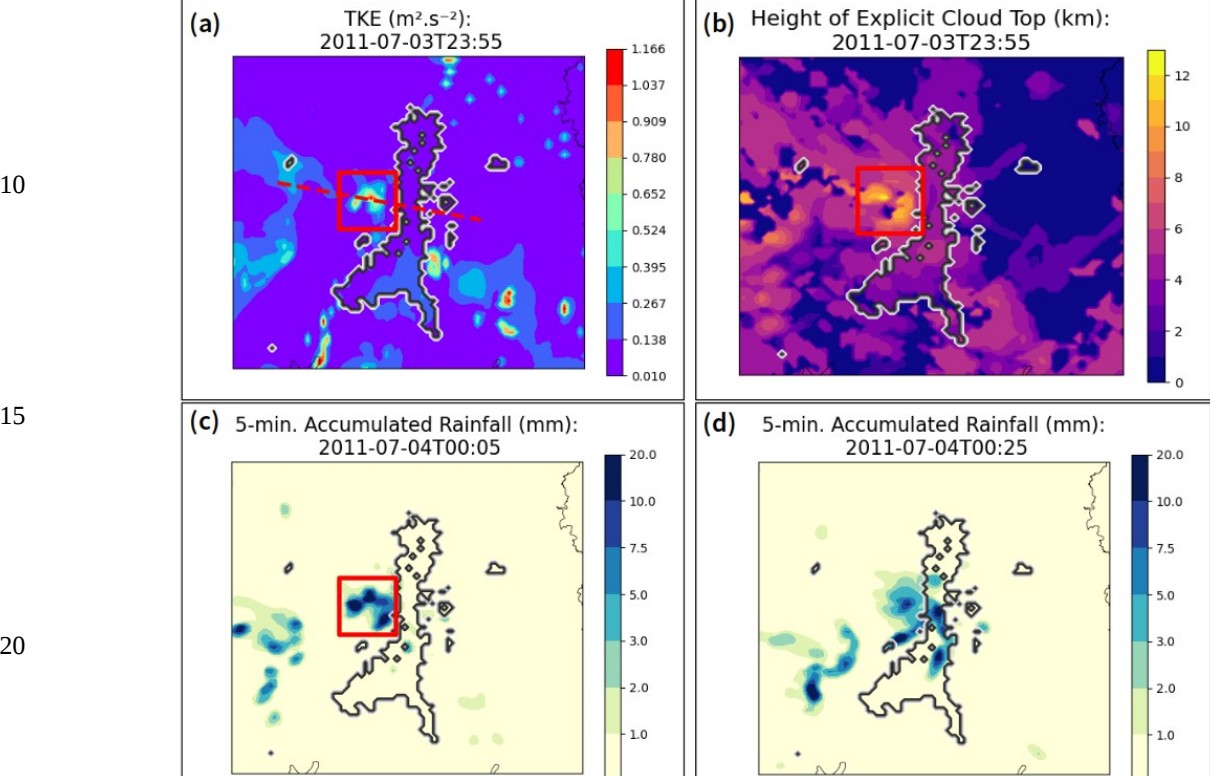

**Figure 8** – Map of TKE in the Kolkata area (a), with the peak of TKE produced by dynamical effect (red square) when the rainfall system reaches the city. The red dashed lines represent the path of the second cross section. The explicit cloud top height at the same time step is shown (b), as well as the 5-minutes accumulated rainfall 10 minutes (c) and 30 minutes (d) after the burst. Panel (c) correspond to the rainfall accumulated between 00:00 and 00:05 on July 4th, and (d) to the rainfall accumulated between 00:20 and 00:25.

To further understand the local wind patterns corresponding to the TKE burst induced by urban terrain, we have analyzed the circulation along a second cross section, this time following the direction of the moderate rainfall system (West to East) and cutting Kolkata perpendicularly (see Fig. 8 (a), dashed lines). We have also plotted the TKE anomalies along this new cross section. The TKE anomalies (Fig. 9 (a)) have been calculated at each vertical level by subtracting the spatial mean of TKE of the level in the Kolkata are from the values of TKE, such as:

$$TKE_{an}(x,y,z) = TKE(x,y,z) - \overline{TKE(z)}$$

Where the last term is equal to the horizontal-mean TKE of level z within the Kolkata area. These anomalies therefore give us information on the values of TKE with respect to their horizontal surroundings, and hence the horizontal propagation of the system.

The analysis confirms that the city acts as a kinetic barrier in the Urban simulation (Fig. 9), forces the surface flow to stop when arriving in the proximity of Kolkata (delimited in black dashed lines), and generates a dynamic production of TKE near the surface (Fig. 9 (b)). This causes the air mass to be lifted, resulting in intense updrafts, which eventually leads to the cloud system developing up until about thirteen kilometers of altitude. Note that the situation at preceding time steps also show the same type of circulation pattern, where the city seems to induce a mechanical lifting. As a sharp contrast to the results of Urban run, the flow remains steady in the No-Urban run, passing through the city. Furthermore, we believe that this barrier effect was enhanced by the geometry of the city. Indeed, Kolkata has an unusually long shape, and the distance from the northern extremity to the southern is about 50 kilometers. Therefore, when the light rain system propagated towards the city in a direction almost exactly perpendicular to the city, the barrier

effect disturbed the surface wind patterns across the whole western boundary of the city, on a distance of several dozens of kilometers. This eventually lead to the TKE burst occurring in the proximity of the middle of this western boundary.

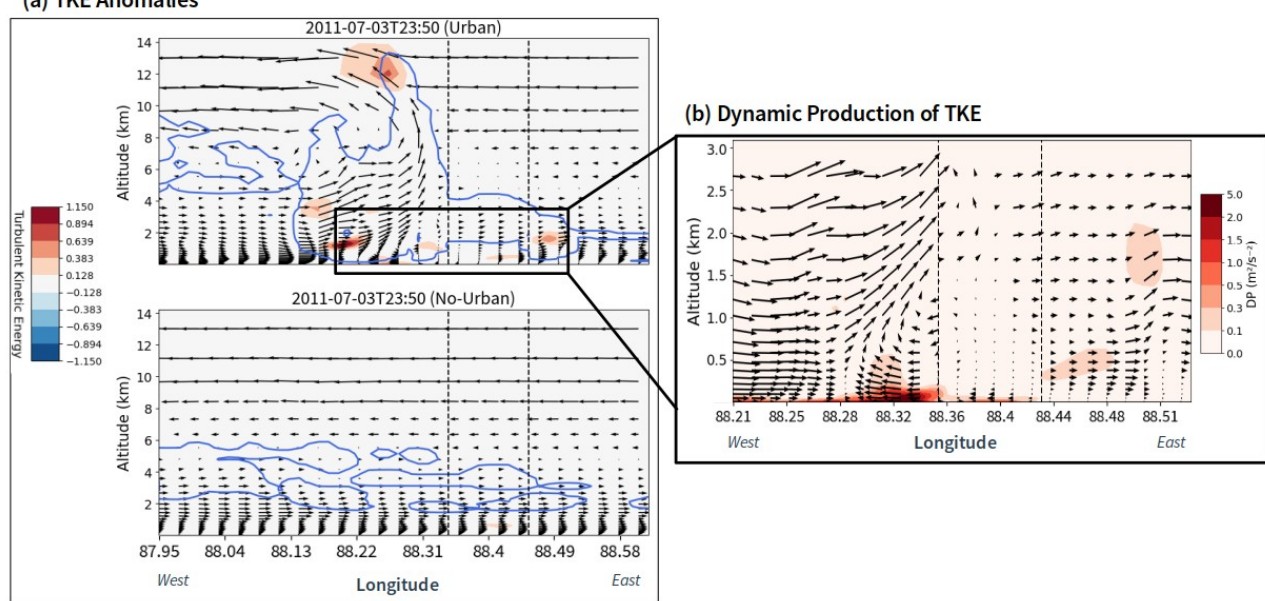

**Figure 9** – Circulations along the West-to-East cross section and TKE Anomalies (a) in the Urban (top panel) and No-Urban (bottom panel) simulations, on July 3$^{rd}$, 23:50. The portion of the cross section where Kolkata is located is indicated in black vertical dashed lines. The blue contour represents the TCW contour (TCW=0.3 g/kg). A zoom in the first three kilometers of atmosphere has been plotted (b), in which the colors represent the dynamic production of TKE,

i.e., TKE induced by wind shear and friction. The values represent the dynamic production of TKE between 23:45 and 23:50 on July 3$^{rd}$.

Although the kinetic barrier effect of the city described here seems to be the direct reason for the storm initiation over Kolkata, the thermodynamical effect might have also contributed to the process. As mentioned above, the TKE have

560 two distinct potential production sources: dynamic and thermal. While the dynamic contribution can only be positive, meaning that the TKE can only be increased by the turbulent modification of the flow, the thermal contribution can take both positive and negative values. This production term is positive when the buoyancy is positive, i.e., when the thermodynamical conditions allow for turbulent mixing in the atmosphere. It takes negative values however, when the atmosphere is thermodynamically stable and the buoyancy is negative. Usually, the buoyancy is positive during the day

when the solar heating at the surface generates convection, and becomes negative at night when the heating stops and the atmosphere stabilizes. However, as discussed in section 3 of this paper, the thermal properties of urban areas create a perturbation in the night time stability that could lead to the buoyancy staying positive during the night. And as expected, this has been revealed by the thermal production of TKE at the city surface during the whole 28 hours derived with frequent outputs (Fig.10). The thermal production is stronger during the day, and intensified by the UHI in the

Urban run. But what is important to note is what happens at the end of the day on July 3$^{rd}$. During the few hours preceding the storm initiation (from 16:00 to midnight), the thermal production of TKE remains positive in the Urban run until the storm starts over Kolkata, therefore keeping the PBL unstable and thus benefiting the convection development. In the No-Urban run, however, this production becomes negative around 16:30 . This shows that the turbulent mixing and convection was still present throughout the night over Kolkata in the Urban run, while the

atmosphere quickly stabilized in the No-Urban run.

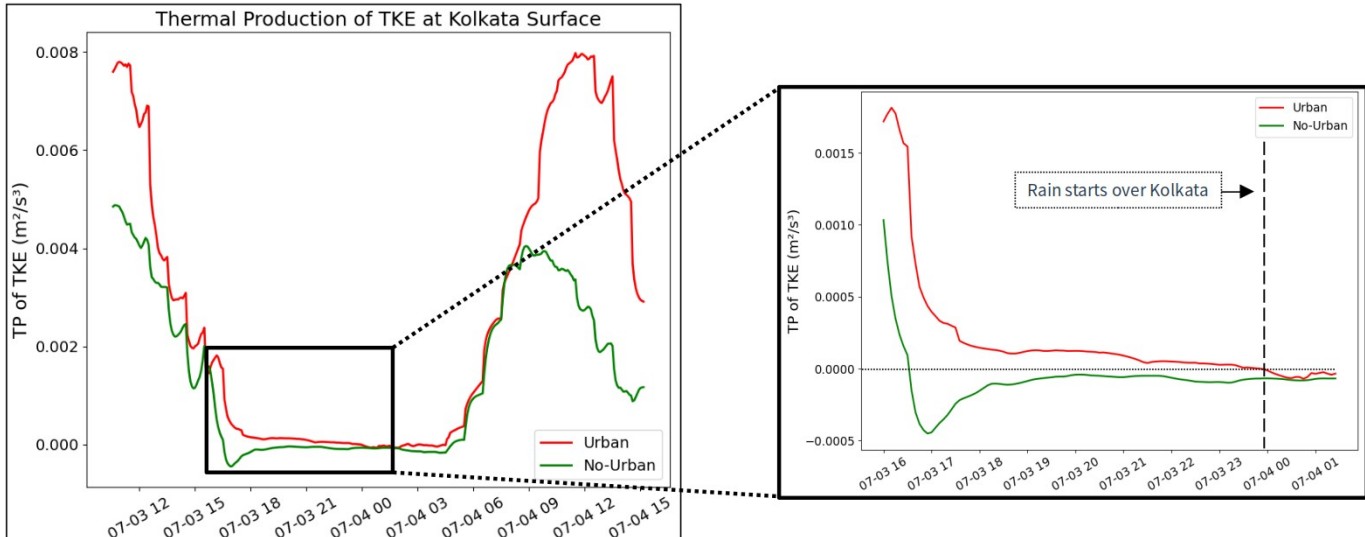

**Figure 10** – Thermal production of TKE at the surface of Kolkata, for Urban (red) and No-Urban (green) simulations. The right panel corresponds to a focus on the night that the storm was initiated above Kolkata. The vertical black dashed line indicates when the rain starts over the city.

The main reason why instability and convection would contribute to storm development is that in a humid environment, the low-level humidity would be lifted from the ground to the upper layers of atmosphere, which would enhance the condensation processes at cloud base. To check if this occurs, we have analyzed the vertical profiles of water vapor mixing ratios in the Kolkata area, for both simulation (Fig. S4). During the few hours prior to the storm, the humidity amounts between 200 and 400 meters of altitude (right above the nocturnal boundary layer) are indeed enhanced in the Urban simulation. The enhanced humidity content could have amplified the condensation process and invigorated the storm.

### 4.4. Rainfall reduction after the storm

There is a fifth peak of rainfall during the first heavy rain period (July 1st to July 5th), which appears to be more intense in the No-Urban run, as indicated by the negative blue peak in Fig. 6 (b). Just like the first three peaks, this intense rainfall is associated with a diurnal convection under favorable synoptic conditions (i.e., low pressure system, which will be discussed in the next section). However, the Urban run appears to have induced a rainfall reduction this time instead of a rainfall enhancement. In order to understand what have caused this rainfall reduction due to urban setting, averaged Convective Available Potential Energy (CAPE) as well as precipitable water in the Kolkata area has been derived for the entire first heavy rainfall period from July 1 – 5 (Fig. 11). CAPE is a measure of the energy available susceptible of being converted into kinetic energy in the clouds in the form of updrafts, and is thus a good metric of the potential of heavy rain events. Precipitable water, on the other hand, represents the available water in the atmosphere for precipitation. It corresponds to the amount of precipitation that would occur if all the water vapor in a vertical column was to condense and precipitate, and is therefore closely linked to the vertically integrated moisture content.    The CAPE results suggest that when the July 3 storm starts to form, the values of averaged CAPE over Kolkata area are nearly identical in both simulations, and the difference between them is almost zero (Fig. 11 (b), first vertical black dashed line). However, after the storm passes over Kolkata (the storm time window is delimited by the

black dashed lines), CAPE is rather low in the Urban run because it has been consumed by the nocturnal storm initiated in the city. Whereas in the No-Urban run, CAPE continues to increase throughout the night, as the precipitation remains light in that case so the energy is still being accumulating. Hence, when the diurnal convective precipitation is initiated the next day after the previous storm, the CAPE is much higher in the No-Urban case than in the Urban case (3288 J/kg in No-Urban against 1877 J/kg in Urban, or 75% higher in No-Urban).

Regarding the precipitable water, similar type of behavior is found (Fig. 11 (c)). At the initiation time of the nighttime storm, the amount of precipitable water in the Kolkata area are identical in both simulations. When the storm starts, we see that the precipitable water in the Kolkata area in the Urban run quickly decreases due to the consumption by the storm. As a result, at the end of the storm and when the precipitation starts on the fourth simulated day, the precipitable water is evidently lower in the Urban than the No-Urban run.

Therefore, the significant rainfall reduction after the urban-initiated nocturnal storm is explained by the consumption of CAPE as well as available moisture by the storm. Nevertheless, the overall rainfall difference between the two runs during the first heavy rain period remains in favor of the Urban run.

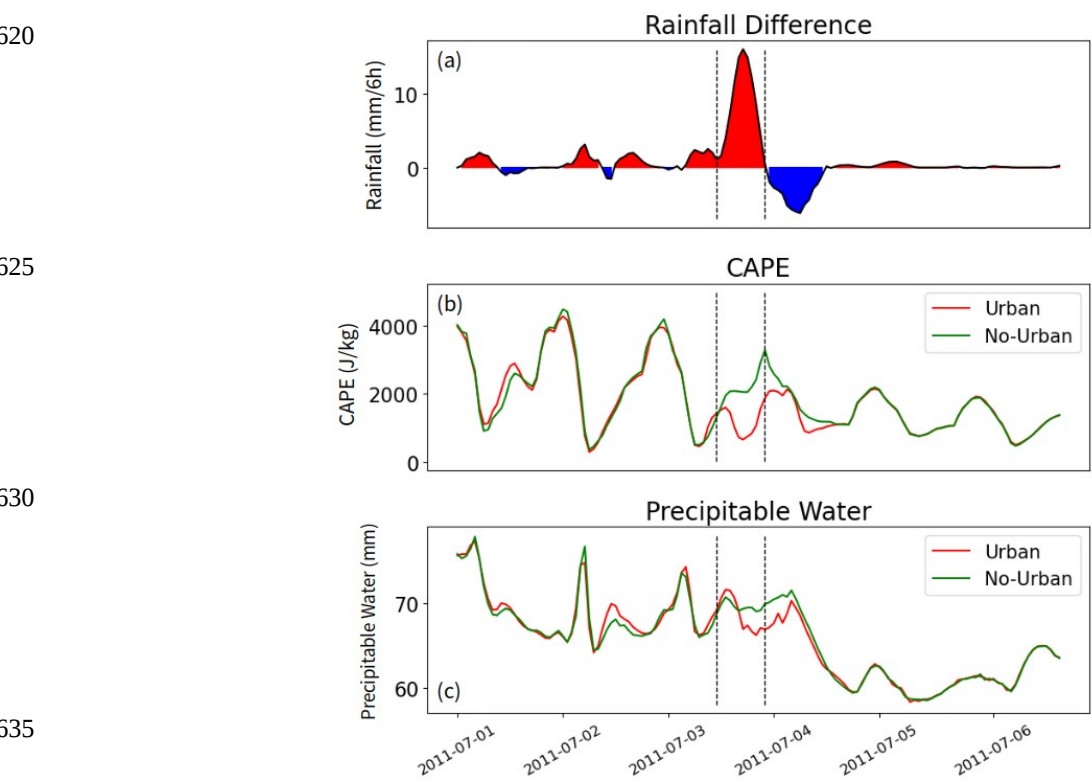

**Figure 11** – Rainfall difference, CAPE and precipitable water time series. The rainfall difference (Urban – No-Urban) during the first perturbation is plotted in (a), the corresponding CAPE and precipitable water time series, calculated in the Kolkata area, in (b) and (c), respectively. The passing of the storm initiated by Kolkata over the area is indicated in vertical dashed lines.

## 5. Discussion on the potential influence of the synoptic-scale conditions

Our analysis indicates that the first and the third heavy rain periods shown in Fig. 6 both are associated with a synoptic scale low pressure system. The low pressure system corresponding to the first heavy rain period appears to be located at the North of India along the Himalayan foothills (Fig 12, (a)), which induces a persistent precipitation in our

domain with an average rainfall rate of 7.2 mm/6h. The low pressure system corresponding to the third period, however, seems to be located in Northeastern India, right above our domain (Fig 12 (b)), producing an even more intense overall precipitation of 9.5 mm/6h on average between July 19th and July 22nd. It is the difference in the location of the associated low pressure system that seems having caused a different rainfall between the two heavy rain periods.

From July 1st to July 5th, the monsoonal winds blow from the Arabian Sea towards what appears to be a trough, located at the Himalayan foothills, before arriving in Northeastern India. Part of the moisture brought by the Northwesterly wind is advected into our domain, and the  convective precipitation systems that we simulated during this period propagate southeastward. On the other hand, from July 19th to July 22nd, the low pressure system is located right over our domain. Most of the moisture is directly advected from the Bay of Bengal, which leads to a more intense

moisture convergence flux in comparison to the first period (Fig. 12 (c)), creating a more humid condition and leading to more intense rainfall. As a comparison, the average moisture convergence flux (indicated in black dashed lines on Fig. 12 (c)) is 0.23 g/m³/s for the first period, while 0.35 g/m³/s for the third one.

   In the three day duration of the last heavy rain period, the rain hardly ever stops in either of the simulations. under such an unstable condition, it is unlikely that the urban area would specifically induce the formation of storms, as the

rain-producing systems are  essentially driven and regulated by the low pressure system. However, during the first period, it is perhaps due to a relatively weak and also distant low pressure system, that sufficient moisture transport from the Northwest of the domain in working with a kinetic perturbation and thermodynamically unstable conditions over the urban area could have initiated the storm in the Urban simulation.

   Note that despite the absence of any 'urban-initiated' storms in the third heavy rain period, rainfall has still been

intensified by the urban area. The most notable difference happened during the night of July 19th, between 20:00 on July 19 – and 8:00 on July 20, the average rain rate was about 8.8 mm/6h in the No-Urban run, while over 13.5 mm/6h in the Urban run, corresponding to a 54% increase caused by the urban land-use. We find that this increase was preceded by an anomalously high UHI, which lead to a surge of surface Moist Static Energy or MSE, a good indicator of the convective activity particularly of the ISM (Fig. S5), implying that the intense UHI has likely caused the maximum

rainfall enhancement over this heavy rain period.  Furthermore, right after the above discussed rainfall peak, we notice an even more intense positive difference peak (Fig. S5 (c)), immediately followed by a negative difference peak. This is simply an indicator of a phase-shift of a rainfall system, both peaks in the Urban and No-Urban run correspond to the same system but initiated a bit earlier in the former simulation. Overall, the rainfall difference is light for this system.

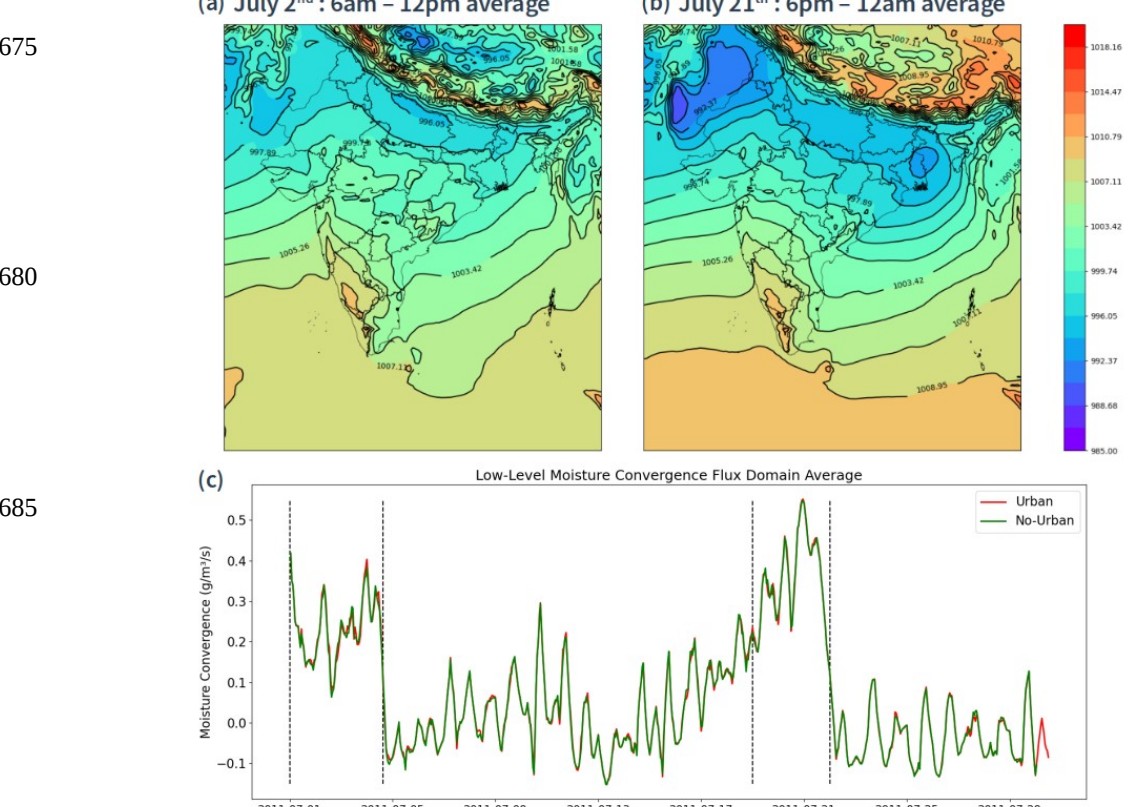


**Figure 12** – Mean sea level pressure (hPa) during the first (a) and third (b) perturbations, derived from the ERA5 Reanalysis dataset. The associated simulated moisture convergence flux in our domain, integrated over the first three kilometers of atmosphere, is shown in (c). The two perturbations are indicated in vertical dashed lines.

## 6. Summary

In this paper, we have assessed the impact of the urban land-use on the precipitation during the Indian summer monsoon in the Kolkata area. By simulating the month of July 2011 with and without the urban land-use, we find that the overall mean rainfall was increased due to urban setting by over 14% in an area of 80 km by 80 km surrounding Kolkata. In general, we find that the enhancement introduced by urban LULC effects is consistent during two heavy rainfall periods corresponding to synoptic low pressure systems positioned in different locations.

The urban heat island effect appears as the most consistent perturbation during the whole modeled month. Its close link with the increase of PBL height suggests that the convection is enhanced by this thermal amplification. While the diurnal convective rainfall (7:00-19:00) enhancement can reach almost 10% in some days, such as the first three modeled days, the overall diurnal rainfall modification is not evident when analyzing the whole month of July 2011. Just like the UHI, the modification of rainfall by the urban area is, however, higher and more persistent during night

time. In particular, the two most intense rainfall modification cases happened during the night, specifically, the night of July 3$^{rd}$ when the urban-initiated storm occurred, and the night of July 19$^{th}$ during the third heavy rain period, which was preceded by an anomalously high UHI. The close relationship between UHI, PBL heights and rainfall enhancement suggests that the thermodynamical effects are the main drivers for rainfall modification on the whole modeled timescale. Our results are in general agreement with several previous studies, which also highlighted the importance of

thermodynamical effects of urban land-use on heavy rainfall in this region (Li et al. 2017; Swain et al. 2023).

Nevertheless, our modeling study has also provided additional insights of the urban kinetic effects on monsoon precipitation, revealed through a modeled storm initiation case over the city of Kolkata, where the dynamical production of turbulent kinetic energy due to urban terrain has played a critical role. In depth analyses on this storm suggest that the reasons behind this urban-initiated storm case include:

1. A sufficient large-scale moisture transport from the Arabian Sea all the way through northeastern India;

2. An additional warm and moist surface flow coming from the Bay of Bengal; and

3. A sudden burst of dynamically produced TKE due to urban terrain at the city border, with a well prepared unstable nocturnal planetary boundary layer due to urban heat island effect.

Specifically regarding the last point, TKE is  greatly enhanced as the surface wind flow decelerates when arriving at

the city, thus increasing wind shear and turbulence. The deep convection was initiated shortly after that, and the heavy rainfall started at the location of the TKE surge. It was also found that the nighttime urban atmosphere was still positively buoyant, thus contributing to this storm initiation. Additionally, we believe that the geometry of the city and the orientation of the initial rainfall system could have influenced this storm generation. The initial system was propagating perpendicularly to the orientation of the city, meaning that the surface winds decelerated across the whole

west border of the city, eventually leading to the TKE burst.

There have been studies showing that thunderstorm formation was more intense around urban areas, and that urbanization might have impacts on storm life cycles and intensities. Here for the first time, we have been able to simulate a case of storm initiation over a city, and thus provide a detailed description of the mechanisms behind this generation. Furthermore, the city-generated storms could be a physical explanation for the observed increase in the
frequency of ISM extreme precipitation events during the past century, which had previously been linked with urbanization. Additional work would need to be done in order to assess the dependency of urban storm formation to external factors such as inter-annual variability or synoptic conditions. Regional climate modeling analyses could also provide confirmation for this hypothesis.


## Author contribution

R.F. and C.W. designed the modeling strategy. R.F. performed and analyzed the simulations. R.F. and C.W. wrote the paper.

## Competing interests

The authors declare that they have no conflict of interest.

## Acknowledgments

This study is supported by L'Agence National de la Recherche (ANR) of France under "Programme d'Investissements
d'Avenir" (ANR-18-MPGA-003 EUROACE).

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
