# Peer review of "Impact of urban land use on mean and heavy rainfall during the Indian Summer Monsoon"

_EGUsphere, 2023_

## Referee Comment (RC2)

**Comments on paper entitled "Impact of urban land use on mean and heavy rainfall during the Indian Summer Monsoon" Atmospheric Chemistry and Physics Journal of egusphere-2023-1445.**

This is an excellent work on urban meteorology and precipitation. It gives a detailed overview of initiation, movement, and mechanism of a storm over both urban and no-urban simulations using Meso-NH model. I am having the following minor corrections before its acceptance in the journal.

1. How have you calculated anomaly of equivalent potential temperature and TKE (Fig. 7 and 9)? Please describe it in detail like how many days or years you have considered for calculating anomaly?
2. Indicate proper equations for calculating both thermal and dynamic TKE.
3. In line number 412, "Therefore, we have conducted an additional set of two….". Please clearly mention the word 'two'.
4. Provide color bar for Fig. 6.
5. Replace the word "four different time steps:" to "three different time steps:" in Fig 7 caption.
6. Provide X-axis label as latitude and longitude values in Fig. 7 and Fig. 9 respectively. It will be easier to understand the figure.
7. Please cite one more recent paper on observational study on urbanization and increased precipitation in the Bhubaneswar region located in Northeastern India by Swain et al. 2023.
   - Swain, M., Nadimpalli, R.R., Mohanty, U.C., Guhathakurta, P., Gupta, A., Kaginalkar, A., Chen, F. and Niyogi, D., (2023) Delay in timing and spatial reorganization of rainfall due to urbanization-analysis over India's smart city Bhubaneswar. *Computational Urban Science*, *3*(1),.8.

---

## Author Comment (AC1)

**Response to the Reviewers' comments**

We truly appreciate all the constructive comments and suggestions made by both reviewers. Following these comments, we have made certain significant revisions and also provided information that would benefit the readers to better understand our methods and results. We also want to clarify a point in the manuscript that was not brought up by the reviewers, that we stated at line 47 of the original manuscript: "The reasons for the existence of this UHI include the decrease of surface albedo as well as the increase in thermal capacity of urban build-up in comparison to vegetation." This sentence is not entirely correct since the urban built-up has a higher thermal conductivity than vegetation, but lower heat capacity. This has been modified in the revised version.

Following the comments of the reviewers, we have made several changes on the figures as well, including:

- Figure 6: The color bar has been added
- Figure 7: The latitude of the points of the cross section have been added, as well as the orientation South / North.
- Figure 9: The longitude of the points of the second cross section have been added, as well as the orientation West / East.
- Figure 11: A third panel representing the time series of the precipitable water averaged in the Kolkata area has been added.
- Figure S2: The name 'Calcutta' has been replaced by 'Kolkata' for consistency.

The following are our point by point responses to each of the comments and suggestions of the reviewers (marked by italic font):

**Response to Reviewer #1:**

*"1. Line 51-52: "Furthermore, vegetation tends to have a cooling effect on its surrounding environment through evapotranspiration and latent heat release,". Line203-205: "As the release of latent heat through evaporation causes a cooling of the atmosphere, the UHI effect is further strengthened by such a reduction of latent heat flux.". These sentences are misleading, evaporation does not release latent heat, but condensation does. Also, the release of latent heat results the heating of the atmosphere."*

**Response:** We thank the reviewer for the comment, as we realize that the term "latent heat release" could induce some confusion. This was corrected in the revised version by stating that the evaporation process does indeed induce a local cooling effect as well as an increase of the latent heat flux from the ground to the atmosphere, as the energy required for phase change is taken from the surroundings and stored within water vapor as latent heat.

*"2. Line 97: the word "characteristic" and "features" are repeating."*

**Response:** This has been corrected in the revised version, only the word "features" was kept.

*"3. Line 161, in subtitle 2.3, "Model comparison with observations". In fact, the model simulated rainfall was compared with the ERA5 reanalysis data in the manuscript. In my opinion, the reanalysis data is not representative of the real rainfall observations, there are some uncertainties when compared to observations. Why the authors used the reanalysis data here instead of the rain gauge observation? What about the uncertainties of the ERA5 rainfall? Line 162, "the model has largely reproduced the evolution of domain-averaged hourly rainfall time series in ERA5 (Fig.*

*2)."This is a subjective and qualitative description. In Fig.2, the model captured the rainfall well only in part of the period, such as the two major heavy rainfall periods. For those days with small magnitude of precipitation, the simulation bias is evident. Especially for the period after heavy rainfall, the model and ERA5 exhibit almost opposite phases. As we all know, accurately simulating rainfall is challenging, the authors may provide some quantitative evaluation and cite previous studies to demonstrate the plausibility of these simulation results."*

**Response:** We thank the reviewer for the comment. Indeed, the model was compared with reanalysis data rather than observations. The mistake in the title of the subsection has been corrected in the revised version.

The reason that we chose to compare our model results with reanalysis data is that ERA5 provides consistent and high resolution data for the whole modeled month over the model domain. While it is not easy to perform a similar comparison using the rain gauge data, not to mention that based on our knowledge the latter data have been used in producing ERA5 reanalysis data. Furthermore, the primary intention of the study is not to simulate the observed rainfall as accurately as possible, but rather to investigate the sensitivity of modeled rainfall to land-use modification. Therefore, a qualitative comparison between the modeled results and reanalysis data appeared satisfactory. We did not expect to simulate the exact rainfall values, but rather insure that the overall rainfall was well represented, which is the case as we simulate the same heavy rainfall periods as the reanalysis data. Details have been added in the revised manuscript.

*"4. In this reviewer's opinion, the inconsistency between the use of a 12-hour clock format in the text (e.g., 2pm, 7am, et.) and a 24-hour clock format in the figures could make reading less convenient."*

**Response:** The reviewer's comment is well received. The text has been accordingly modified in the revised manuscript to match the figures under a 24-hour clock format.

*"5. Line 216-217: it seems that the UHI reached the maximum at 9 pm in Figure 3b. Line 219: the UHI intensity starts to decrease after 9 pm in the Figure 3, I guess the author means that the UHI starts to rapidly decrease after midnight."*

**Response:** On figure 3b, the value of the UHI is indeed maximum at 7 pm as indicated in the manuscript, however, the values at 7 pm and 9 pm are almost identical (0.654 K at 7pm versus 0.650 at 9pm). The reviewer is right that we did mean that the UHI was decreasing very slowly until midnight, and more rapidly after. This has been clarified in the revised manuscript.

*"6. Line 226-227: Indeed, the PBL decreases quickly in both Urban and non-Urban runs, while remains relatively higher in the Urban run."*

**Response:** This point was well understood by the reviewer.

*"7. Line 238: "4 and 5 pm"->"5 and 4 pm"."*

**Response:** Corrected.

*"8. Line 242: "This result is another indicator of the residual instability of the nighttime urban boundary layer.", what does the sentence mean?"*

**Response:** We acknowledge that a further clarification is needed here. Our point was that in subsection 3.1, we have shown that the nighttime boundary layer remained unstable and the convection was still active at night, as indicated by the PBL being more than twice as high in the Urban run. Then in section 3.2, the diurnal cycles of rainfall showed that the enhancement of

rainfall was more evident at nighttime, which is a confirmation that the convective activity is still present at night in the Urban run, and indicates instability in the nighttime urban boundary layer. This has been clarified in the revised manuscript.

*"9. Line 264-265: " This rainfall enhancement by the urban setting appears to be even more intense over the city itself". However, from Fig. S1, it seems that the urban effect on rainfall enhancement is more pronounced in the surrounding areas rather than the urban itself. Is there any possible explanation?"*

**Response:** We understand the reviewer's confusion, as when looking at Fig. S1, some intense rainfall spots are located around the urban area. However, what is indicated in the manuscript is also true. The average rainfall values in the whole Kolkata area are 2.630 mm/6h in the Urban run and 2.299 mm/6h in the No-Urban run, or an average increase of 14.4%. Furthermore, when only including the grid cells located over Kolkata in the same figure, the average rainfall values become 2.712 mm/6h and 2.208 mm/6h in the Urban and No-Urban runs, respectively, or a 22.8% increase as indicated in the following sentence in the manuscript.

Regarding the fact that some intense spots are located around the urban area, it is not surprising that the maximum rainfall enhancement happens outside of the city border, as the location of the modification is intricately linked with the local circulation. An additional analysis on wind patterns would be required to explain the exact location of the spots.

*"10. Line 287: There is inconsistent city name, "Kolkata area" in manuscript and"Calcutta Area" in Figure S2."*

**Response:** We thank the reviewer's for pointing out the inconsistent usage. "Calcutta" has been replaced by "Kolkata" throughout the revised manuscript as well as in the supplementary material.

*"11. In Section 4, the first two rainfall peaks exhibit less differences between Urban and non-Urban runs, but the third peak shows evident difference. Is this phenomenon solely due to the first two rainfall events occurred during the daytime? Urban effects exist both during the day and at night. Also, the barrier effect of urban dose not disappear during the daytime. Why are the impacts of urban effects on the rainfall associated with the first two rainfall peaks almost negligible, is there further in-depth reasons for this? For example, the urban area blocks the evaporation and transport of water vapor, compared to non-urban areas represented by grasslands in this study. When the path of a storm is parallel to the orientation of a city, especially in the case of the strip-shaped city described in this paper, the barrier effect of the city may be minimal. Additionally, it can be noticed that the rainfall is presented in a regional average manner, while the spatial scale of the impacts generated by urban effects is often relatively small. Within a larger spatial domain, is there a possibility that the rainfall enhancement is offset by an urban effect induced rainfall reduction? In addition, as the geographical specifics of the Indian Summer Monsoon, it is needed to highlight the influences of the Tibetan Plateau and the Indian Ocean Sea on clouds, aerosols, and circulation if possible. Recent studies have focused on these issues (such as, Wang et al., 2022, JGR; Liu et al., 2022, ESR). The authors may add a little discussions."*

**Response:** There are several very interesting points raised by the reviewer in this comment.
- First, regarding the first few rainfall peaks: the first three simulated peaks of rainfall indeed show less difference between the two runs than the fourth one. However, such a difference is still non-negligible, as the rainfall maximum for all three peaks are about 10% higher in the Urban run. Those first three peaks of rain all happened at the end of each of the first three days (around 6 ~

7pm). This is typical of diurnal convective rainfall: the rainfall is intense during the day when the convection is active, which, given the sufficient large-scale moisture supply from the Arabian Sea, produces the intense precipitation of the first heavy rain period. As we have shown in section 3, the convection is more intense in the Urban run due to the thermodynamical effect of the UHI, which explains why these three convective rainfall peaks are 10% higher than in the No-Urban. Nevertheless, these peak being "only" 10% higher can be explained by the fact that during the day, the rainfall processes are essentially driven by the incoming solar radiation as well as moisture supplies. The urban area only appears as a 'perturbation' during the day in our simulation.

This is why the third modeled night differs from the beginning of the simulation. When looking at the fourth rainfall peak (i.e., the third night), it is evident that the processes involved are different, because the rainfall is close to zero in the No-Urban run and as intense as the first three peaks in the Urban run. Instead of being just amplified by about 10%, the rainfall is more than 3 times more intense (~ 240% increase over the whole night). This is why we talk about rainfall 'enhancement' in the former case, and rainfall 'initiation' in the latter. Note that Section 4 of the paper is dedicated to explaining the physical processes occurring during this fourth rainfall peak. What we show in Section 4, is that this initiation is primarily due to the urban barrier effect, while the nighttime thermodynamical perturbation appears as a contributing factor. Note also that during the first two nights, the rainfall stopped completely in both simulations, that means that even in the Urban run and despite the nighttime instability, the perturbation of the city was not strong enough to produce nighttime precipitation. All these suggests that it's a combination of factors (TKE production, thermodynamical instability, moisture availability) that lead to the storm initiation on that 3rd night.

To sum up on this comment, the reviewer is right that the urban effects are active during both the day and night, although with different magnitudes. Sometimes, the urban effect can simply slightly enhance the rainfall but with the right combination of factors, it can also induce the formation of heavy rainfall. We have added additional sentences in the revised version in order to clarify these processes.

- Secondly, the reviewer makes an important comment on the effect of the orientation of the city. We believe that this point is crucial in the discussion, and while we mention it in the Summary section of the paper, we have made modifications in the revised manuscript to further emphasize on this in the main text. This geometric consideration might be an essential reason as to why storm initiation happens at only one occasion during the whole modeled month. Kolkata has a particularly long shape, and the TKE burst happened when the surface wind was perpendicular to the city, thus maximizing the urban barrier effect. Therefore, the reviewer is right in the way that the barrier effect of the city might not disappear during daytime, however, it was maximum during this particular event as it even induced the formation of the heavy rain event.

- The last point addressed by the reviewer is also a crucial one. Indeed, the precipitation results presented in the paper are for the most part calculated in an area of 80km by 80km around the city, noted as the 'Kolkata area'. We believe that the reviewer's concern comes from the fact that despite an enhancement in the immediate proximity of the city, there might be reduction of the rainfall further away. However, we did not see any area within our domain that showed significant or systematic rainfall reduction. That being said, we chose to calculate the precipitation results in a relatively small analysis domain like the Kolkata area because, as the reviewer also acknowledged, the urban effects are indeed quite localized. That means that despite the significant differences that we have shown in the Kolkata area, such differences remain fairly light when looking at the rainfall over the whole domain (which covers an area of about 400 km by 400 km). This is not due to a

rainfall reduction that would offset the enhancement in the Kolkata area, but rather due to the absence of significant differences anywhere else in the domain, thus dimming the effect of Kolkata on the whole domain rainfall. Furthermore, the choice of the size of the 'Kolkata area' was made following the meta-analysis from Liu and Niyogi (2019), in which they showed that most of the rainfall modifications happened under 60 km away from the city center (we chose 80 km to ensure that we have most of the effects). Following the reviewer's comment, we have added a comment about the localized nature of the results in the revised manuscript.

*"12. In Section4.4, a fifth peak of rainfall during the first heavy rain period (July 1st to July 5th), which appears more intense rainfall in the non-Urban run. The authors analyzed the corresponding CAPE time series and concluded that the rainfall during the fourth peak consumed the CAPE energy, resulting in the opposite rainfall differences during the fifth peak. The CAPE in the Urban run is indeed lower than that in the non-Urban run during the rainfall processes of the fourth and fifth peak. Nevertheless, CAPE is not the only factor, as the CAPE corresponding to the fifth peak is not significantly lower in Urban run compared to non-Urban run. Additionally, during the rainfall on July 1st-2nd, high CAPE in the Urban run even corresponded to low precipitation in the Urban run. From a mechanistic perspective, the consumption of CAPE does reduce the energy available to trigger convection. However, could there be additional reasons, such as the fourth peak's consumption of the available precipitation leading to insufficient moisture supply for rainfall in the fifth peak?"*

**Response:** We thank the reviewer for this last comment, which we find to be particularly relevant. It is true that precipitation processes are complex and CAPE is not the only factor influencing the results. We chose to explain the negative rainfall difference of the fifth peak by displaying the CAPE time series because they clearly highlighted the storm passing over the city in the Urban run and the consumption of the energy. We still argue that when the convective rainfall starts on the fifth day, the CAPE is 75% higher in the No-Urban run, and thus we find the difference to be non-negligible. Nevertheless, the available moisture is another key factor to be considered, and we realize that the reader might need this information for a better understanding.

Therefore, we have added in the revised manuscript a third panel in Fig. 11, displaying the time series of precipitable water (representing the vertically integrated moisture), averaged in the Kolkata area. These time series show that in addition to consuming the CAPE, the nighttime storm over Kolkata also consumes the available moisture. We mention in the revised manuscript that this could be a contributing factor to the rainfall reduction of the fifth peak.

**Response to Reviewer #2:**

*"1. How have you calculated anomaly of equivalent potential temperature and TKE (Fig. 7 and 9)? Please describe it in detail like how many days or years you have considered for calculating anomaly?"*

**Response:** We thank the reviewer for the comment, as we realize it needs further clarification. Both anomalies mentioned are actually spatial and not temporal anomalies (i.e., calculated by subtracting spatial means instead of temporal ones). However, they have been calculated differently.

- Regarding the equivalent potential temperature anomalies in Figure 7, they are calculated at each grid of the cross section by subtracting the vertically averaged $\theta_e$ value in the whole column. Therefore, the anomalies give information on the vertical profiles of $\theta_e$ at each time step and for each grid of the cross section, which is useful for stability assessment. This was explained in the

main text but has now been added in the figure caption in the revised manuscript, for clarity.

- Regarding the TKE anomalies in Figure 9, they have been calculated at each vertical level by subtracting the spatial mean of TKE of the level in the Kolkata area from the values of TKE, such as:

$$TKE_{an}(x,y,z) = TKE(x,y,z) - \overline{TKE(z)}$$

Where the last term is equal to the TKE averaged within the Kolkata area in the horizontal directions (calculated at each vertical level). These anomalies therefore give us information on the values of TKE with respect to their horizontal surroundings, and hence the horizontal propagation of the system. This explanation has also been added in the revised manuscript.

*"2. Indicate proper equations for calculating both thermal and dynamic TKE."*
**Response:** The equation used to calculate the TKE in model has been added in the main text, and the terms corresponding to the thermal and dynamic contributions have been specified.

*"3. In line number 412, "Therefore, we have conducted an additional set of two….". Please clearly mention the word 'two'."*
**Response:** We have modified the sentence as: "...an additional set of two 28-hour (i.e., Urban and No-Urban, respectively) simulations…".

*"4. Provide color bar for Fig. 6."*
**Response:** Done.

*"5. Replace the word "four different time steps:" to "three different time steps:" in Fig 7 caption"*
**Response:** Done.

*"6. Provide X-axis label as latitude and longitude values in Fig. 7 and Fig. 9 respectively. It will be easier to understand the figure."*
**Response:** We have added the latitude and longitude values in Fig.7 and Fig.9, respectively. In addition, the orientation of the cross section is now detailed with the legends North/South/East/West.

*"7. Please cite one more recent paper on observational study on urbanization and increased precipitation in the Bhubaneswar region located in Northeastern India by Swain et al. 2023. • Swain, M., Nadimpalli, R.R., Mohanty, U.C., Guhathakurta, P., Gupta, A., Kaginalkar, A., Chen, F. and Niyogi, D., (2023) Delay in timing and spatial reorganization of rainfall due to urbanization-analysis over India's smart city Bhubaneswar. Computational Urban Science, 3(1),.8"*
**Response:** The suggested paper has been cited in the Introduction of the revised manuscript.

---

## Author Response (AR2)

**Response to the Editor's comments**

Following the editor's last two comments, we have made minor revisions in the final revised manuscript. Notably, we have modified a sentence in the manuscript that could be subject to interpretation and could be perceived as inaccurate, and also added a brief discussion on the influence of large scale features.

Here are the two minor revisions mentioned by the editor:

"*In the revised manuscript Line 277: "The analysis of PBL heights showed that the convection was still present at night in the Urban run, caused by the important nighttime UHI". In this reviewer's opinion, this statement is unfair, as there is no evidence demonstrates a relationship between convection and relatively higher nocturnal PBL. In fact, the nocturnal is generally stable stratification, like its height is around 300 m in present study. The authors may cite some researches to clarify this point, i.e., the higher nocturnal PBL caused by UHI can maintain the convection. Otherwise, the sentence would be better to "The nighttime UHI induced higher nocturnal PBL indicates less atmospheric stability in the Urban run, which favors the possibility of rainfall occurrence.""*

**Response:** Indeed, even in the Urban run, we found that the atmosphere appears to stabilize at nighttime, as we see a decrease in the PBL height. Nevertheless, it's rather the fact that the Urban PBL is twice as high as the No-Urban PBL that suggests that the atmosphere remains relatively less stable in the Urban run. In this sense, we realize that the sentence mentioned by the editor might be slightly misunderstood, and it therefore has been modified by the following sentence in the revised manuscript:

"In the previous subsection, we found that the nighttime UHI induced higher nocturnal PBL, which indicates less atmospheric stability in the Urban run and favors the possibility of rainfall occurrence."

"*To make more complete article structure and research background, my opinion remains that the author may add some discussions regarding the influences of Tibetan Plateau and the Indian Ocean on India summer Monsoon.*"

**Response:** We agree that it is important to remind the reader of the broader context of the study, the Indian monsoon being a complex large-scale system. We have added a paragraph in the Introduction of the revised manuscript stating that urbanization appears as a local perturbation interacting with such larger scale dynamic factors mentioned by the editor. The following paragraph has been added:

"It is worth noting that urbanization appears as a local factor interacting with larger scale dynamic factors. For example, the Indian monsoon is known to be highly influenced by geographic particularities including orography on the West coast, but also in the Northern part of the country, where the Tibetan Plateau can influence aerosol concentrations, circulation, and precipitation (Liu et al., 2022). Thermodynamic considerations over the Indian Ocean can also explain rainfall sensitivity to large scale conditions, especially Northern India which is directly affected by the Bay of Bengal (e.g., Sheehan et al., 2023). Urbanization thus appears as a local perturbation with feedback to these large scale features."